# Mortality from Homicides in Slums in the City of Belo Horizonte, Brazil: An Evaluation of the Impact of a Re-Urbanization Project

**DOI:** 10.3390/ijerph16010154

**Published:** 2019-01-08

**Authors:** Maria Angélica de Salles Dias, Amélia Augusta de Lima Friche, Sueli Aparecida Mingoti, Dário Alves da Silva Costa, Amanda Cristina de Souza Andrade, Fernando Márcio Freire, Veneza Berenice de Oliveira, Waleska Teixeira Caiaffa

**Affiliations:** 1Observatory for Urban Health in Belo Horizonte (OSUBH), School of Medicine, Federal University of Minas Gerais (UFMG), Belo Horizonte 30130100, Brazil; gutafriche@gmail.com (A.A.d.L.F.); darioalves_sc@yahoo.com.br (D.A.d.S.C.); amandasouza_est@yahoo.com.br (A.C.d.S.A.); venezaberenice@gmail.com (V.B.d.O.); caiaffa.waleska@gmail.com (W.T.C.); 2Public Health Post-Graduation Program, School of Medicine, Federal University of Minas Gerais (UFMG), Belo Horizonte 30130100, Brazil; 3Department of Statistics, Institute of Exact Sciences, Federal University of Minas Gerais (UFMG), Belo Horizonte 30130100, Brazil; suelimngt@gmail.com; 4Department of Public Health, Federal University of Mato Grosso (UFMT), Belo Horizonte 30130100, Brazil; 5Department of TI, Belo Horizonte City Hall, Belo Horizonte 30130100, Brazil; ferna@pbh.gov.br

**Keywords:** slum upgrading, urbanization, housing, homicides, urban determinants, poverty areas, slums, health impact evaluation, social determinants of health, health in all policies

## Abstract

*Background*: Homicide rates in Brazil are among the highest worldwide. Although not exclusive to large Brazilian cities, homicides find their most important determinants in cities’ slums. In the last decade, an urban renewal process has been initiated in the city of Belo Horizonte, in Brazil. Named Vila Viva project, it includes structuring urban interventions such as urban renewal, social development actions and land regularization in the slums of the city. This study evaluates the project’s effect on homicide rates according to time and interventions. *Methods*: Homicide rates were analyzed comparing five slums with interventions (S1–S5) to five grouped non-intervened slums (S0), with similar socioeconomic characteristics from 2002 to 2012. Poisson regression model estimates the effect of time of observation and the effect of time of exposure (in years) to a completed intervention, besides the overall risk ratio (RR). *Results*: Using the time of observation in years, homicide rates decreased in the studied period and even more if considered cumulative time of exposure to a completed intervention for S1, S2, S3 and S4, but not for S5. *Conclusions*: Although the results of the effect of the interventions are not repeated in all slums, a downward trend in homicide rates has been found, which is connected to the interventions. New approaches could be necessary in order to verify the nexus between slum renewal projects and the reduction of homicide rates.

## 1. Background

Latin America has been considered the most urbanized region of the global South, with the highest rates of urbanization, poverty, social exclusion and violence [1]. Two characteristics have left a footprint in the urban violence scenario in this region: the high proportion of deaths and the overrepresentation of the young population segment between age 12 and 29, mostly male—both as victims and as perpetrators, living in vulnerable areas of the cities—villas, slums or shantytowns [2].

Latin America has presented one of the largest worldwide increases in homicide rates between 2000 and 2010, as well established in the academic literature and reports from international agencies [3,4,5]. Brazil, as ascertained by the most recent report, reached the highest absolute historical mark of 62,517 homicides, in 2016, with a rate of 30.3 deaths per 100,000 inhabitants, 30 times higher than European rates. In the last ten years alone, 553,000 people have lost their lives due to intentional violence in the country [2].

If on one hand, violence has become increasingly urban and Brazilian cities are the scenario of what is called ‘civic conflict’ [6], on the other hand, cities are also sites of ‘generative civic engagement’ and innovative formats for social mobilization, political participation and creative governance [7]. 

This work presents the example of Belo Horizonte city, a large urban center where the space organization logic is not different from any megacity. Belo Horizonte is characterized by deficient living conditions and the emergence of significant and unjust social issues, such as homicides [8]. The average homicide mortality rate in the city was 25.9/100,000 inhabitants, similar to other Brazilian capitals in 2015, being three to seven times higher in its slums [5,9,10,11].

Driven by a democratic and popular government and a process of participatory governance, in 2002, the Belo Horizonte City Hall (PBH, in Portuguese) started an extensive project for the re-urbanization of its slums. This process includes participatory budget forums, sectoral and cross-sectoral councils of urban and social policies, and participatory forums like those of Vila Viva Project, with similar models to what happens in other Brazilian cities [12,13,14,15,16].

Defining slums as urban spaces in precarious living conditions, insufficient provision of basic services, spontaneous occupation, with predominantly self-built and low-quality buildings, insufficient space, inadequate water and sanitation supply and absence of secure land ownership [15,17,18] the Vila-Viva Project (PVV, in Portuguese) has been considered one of the most encompassing Brazilian re-urbanization models [15,19]. PVV was created in order to include slums in the formal city, legalize them (in the sense of legalizing the land and transferring property ownership), and improve living conditions and their residents’ quality of life, so PVV’s aims fulfill the conceptual framework of interventions in the field of urban health for urban disparities [15]. 

For such urban disadvantage contexts and their avoidable and unjust consequences for the population, the urban and health conferences, studies and revisions, government projects, international organizations and social movements recommend that public urban and social policies be intersectoral, structuring, and related to re-urbanization of townships and slums [20,21,22,23,24].

Galvanized by the devastating incidence of homicides and other health hazards in slums [17], the Observatory for Urban Health in Belo Horizonte (OSUBH, in Portuguese) from the School of Medicine of the Federal University of Minas Gerais (UFMG), in partnership with the PBH, the Oswaldo Cruz Foundation and the Brasilian Ministry of Health set a case study aimed at building an evaluation model for re-urbanization policies in slums and their impact on health. Given the still insufficient national literature on this subject, the ravaging homicide rates were elected in this study as the most significant event to evaluate the effects of re-urbanization policies in the slums.

The international literature presents some conceptual models, different projects of urban interventions, as well as proposals for evaluations of urban revitalization policies that are associated with the reduction in criminality and other health indicators [16,25,26,27,28,29,30,31,32,33,34]. Nevertheless, evaluation experience and even intervention projects are still scarce in Brazil.

Therefore, the purpose of this article, as a part of the larger study called BH-VIVA-OSUBH, is to describe and evaluate the impact of Vila-Viva Project’s interventions on mortality rates due to homicides. Similar slums with and without urban interventions were compared, as a case study model [35].

## 2. Methods

### 2.1. Design, Intervention, Units of Analysis, Data Source and Data Organization

#### 2.1.1. Design

A non-concurrent quasi-experimental study design was used, describing and analyzing homicide rates overtime, in slums with and without Vila-Viva Project’s (PVV) interventions. The study ranged from 2002 to 2012, following the intervention schedule for each studied slum in Belo Horizonte, capital of the state of Minas Gerais, with a population of 2,375,151. Approximately 460,000 people (19% of the population) reside in 186 slums, which account for 5% of the city’s area [15,35].

#### 2.1.2. Background of the Intervention

Vila-Viva Project’s was implemented in 2005 by the Urbanization Company of Belo Horizonte (URBEL, in Portuguese). URBEL has been responsible for the coordination of a program focused on the re-urbanization of slums and precarious settlements, seeking to guarantee access to goods, services and urban structure [15,35].

This is a city policy of social inclusion with integrated urbanization actions for social development and regularization of informal urban settlements, in addition to structural urban interventions. The program is centered on improving residents’ living conditions through urban renovation and environmental recovery, land ownership regularization and legalization, and socio-economic development. It was initially financed by the Inter-American Development Bank (IDB), the National Bank for Economic and Social Development (BNDES in Portuguese), the Ministry of Cities, the Caixa Econômica Federal (CEF) bank and later on by the federal government’s Growth Acceleration Program (PAC in Portuguese) [15,35].

Elaborated with the involvement of local community dwellers’ in each of the slum to be intervened, PVV includes the following steps: assessment and planning, intervention and participatory evaluation. The first step, called Global Specific Plan (PGE in Portuguese), is an in-depth assessment of the social, economic, physical, environmental and land ownership situation.

The diagnosis is carried out by URBEL technicians in the urban and social field, with community participation. The purpose is to diagnose, in detail, the urban-environmental, socio-economic and legal scenario that addresses and allows the analysis of the following items: (1) degree of housing consolidation, based on the concentration of residents per household and the characteristics of the buildings; (2) degree of consolidation of the road system, identifying the physical characteristics and organization of local access; (3) degree of water insalubrity, from the identification of critical drainage areas and the characteristics of local basic sanitation; (4) degree of geological-geotechnical consolidation, relating the types and levels of geotechnical hazards; (5) conditioning and restrictive characteristics of the occupation, through legal analysis and land ownership analysis and, (6) identification of the local social context, the issues experienced by the community and their access to several social policies (with their difficulties and virtues) [15,36]. 

After diagnosing the issues, the next steps in the PGE are to propose interventions and to analyze financial costs. The PGE is the main planning tool of Vila Viva’s Project and it is used to raise funds from the federal government and national and international financial institutions [15,35,36].

As a result of the PGE, URBEL and PBH have agreed on four groups of interventions, which were included in the City’s Master Plan:(1)Urban renewal, comprised of building new streets with improved accessibility to the road system, water and sewage drainage, public lighting and prevention and elimination of geological risk areas for landslide and flooding;(2)Housing construction, consisting of two or three-bedroom apartments;(3)Social development projects, which include: the construction of public facilities, in order to provide health, educational and social assistance; the construction of recreational and cultural areas, sports facilities, parks and green spaces, aiming to achieve environmental recovery and leisure; job and income generation programs as well as projects to foster participatory mechanisms in the community;(4)Regularization of land ownership providing title deeds.

PGE has a specific section regarding the dwellers’ removal as a consequence of interventions in geological risk areas or in areas that will be needed for the works. To the removed dwellers, two options were offered: either to be relocated into new buildings or houses within the slum or to be relocated to houses acquired in any selected city by the dweller within a pre-defined cost. Up to the period studied, removals to other cities were almost non-existent and relocations within the slums did not exceed 25%.

#### 2.1.3. Units of Analysis: The Studied Slums

The choice of slums in this study was based on their history and time of occupation, sociodemographic indicators (sex, age, race, and residents’ income), percentage of residents in census tracts classified according to a georeferenced compound index of social vulnerability (IVS), produced by the Department of Health of Belo Horizonte, as well as social and urban characteristics and issues, such as violence, identified in the PGE diagnosis stage [15,36]. 

Five slums (totalizing 20% of all slum inhabitants in Belo Horizonte) that had interventions were elected by URBEL together with the BH-VIVA-OSUBH team to be part of the study. They were: (S1) Serra; (S2) Morro das Pedras; (S3) Vila São José; (S4) Pedreira Prado Lopes and, (S5) Vila São Tomaz. The comparison group was constituted of five non-intervened slums (S0), composed of Vila Santa Lúcia, Vila Ventosa, Vila Cabana, Vista Alegre and Felicidade, totalizing 18.2% of all slum inhabitants in the city. They presented similar characteristics to the intervention group (Table 1). 

#### 2.1.4. Data Source and Organization

Slum data was collected through a participatory process carried out by the PGE and systematized by the BH-VIVA-OSUBH team’s extensive documentary reading and analysis of the PGE documents in the regional and central archives of URBEL [36]. Supplementary Information of demographic indicators, as well as a compound index of social vulnerability were extracted, respectively, from the IBGE 2000 and 2010 censuses [37] and the City Health Department. 

All information was georeferenced by census tract. In partnership with URBEL and with all possible information at hand, we selected slums according to the following criteria: slums with interventions scheduled to take place at different times (for current and future impact comparisons); slums with broader and more robust interventions; slums with and without interventions that besides being vast in their area and population size, were also situated near upscale neighborhoods; slums situated in less wealthy regions. Besides that, all slums selected had historical backgrounds in the city. We also chose those (intervened and non-intervened) with more severe and complex social and habitational issues, according to Table 1, with similar characteristics. The slums and their location in the city are presented in the map as Figure 1.

All information related to the intervention was collected at URBEL, based on the address, type, scope and schedule, with interventions starting and ending (if concluded) dates. Interventions were mapped over time for each slum, and georeferenced in the census tract. The Intersect Operator Mapinfo software version 8.5 was used to integrate data to the census tract.

Figure 2 shows the georeferenced interventions in one of slum categorized by typology. For analytical purposes, the observation period of this study started in 2002 and was censored in 2012.

The interventions started from 2005 in S1 to 2011 in S5, varying either duration or the cumulative time of exposure. Only S1 had 100% of the interventions finalized in 2011, and therefore contributed with one year of post-intervention analyses with all works completed. For the other four intervened slums, works were still ongoing by the year 2012, and the percentage of conclusion varied from 17 to 90.3%. Table 2 displays detailed characteristics of the built intervention (housing and urban renewal) by slum, including time of initiation, execution and conclusion, as well as total costs. Costs were higher in S1 and S3 because of the size of interventions and connections with their respective surroundings. S5 received less financial input since interventions began at the end of the studied period.

### 2.2. Variables

#### 2.2.1. Response Variable

The homicide rate (per 100,000 inhabitants) per year and per slum was calculated based on the number of homicides divided by the population in the same year. The average rate for the period was calculated dividing the total number of homicides (2002–2012) by the population in the middle of the period (2007) and divided by the 11 years of the period, for each slum or groups with and without interventions.Homicide data were extracted from the Mortality Information System of the Brasilian Ministry of Health and the Belo Horizonte City Health Department from 2002 to 2012. All ICD-10 deaths (International Classification of Diseases) coded as aggressions—X85.0 to Y09.9—were included and georeferenced according to the victim’s residence address in the census tract. The population count was obtained from the censuses 2000 and 2010 of the Brazilian Institute of Geography and Statistics (IBGE in Portuguese) and the population estimated between the census years (2007) [37].

#### 2.2.2. Explanatory Variables

(a)Intervened and non-intervened slums: five intervened slums were denominated S1 to S5; five non-intervened slums were grouped into S0;(b)Time of observation (years: 2002 to 2012);(c)Time of exposure to completed intervention (years), calculated as the sum of all concluded intervention time intervals in a given year divided by the number of all interventions finalized until that year, for each year and in each intervened slum.

### 2.3. Data Analysis

Homicide rates per 100,000 inhabitants were calculated for each intervened slum (S1–S5) and compared with the non-intervened group (S0). Risk ratios were calculated: (1) to estimate the effects of time of observation (years: 2002 to 2012) and cumulative time of exposure to completed intervention (years); (2) to compare intervened and non-intervened slums, using as reference non-intervened slums. Poisson regression model was used to estimate risk ratios (RR) with a 95% confidence interval.

### 2.4. Ethics Approval

The project was approved by the UFMG Research Ethical Committee and by the Belo Horizonte City Health Department (CAAE: 11548913.3.0000.5149).

## 3. Results

Homicide rates in intervened slums ranged from 72.6 to 161.9/100,000 inhabitants and in non-intervened from 65.1 to 141.2/100,000 inhabitants, from 2002 to 2012. Cumulative time of exposure in intervened slums in 2012 ranged from 3.7 to 0.5 years. Figure 2 shows dropping rates in slums S1, S3 and S4 over time, even before the interventions but an increased protection mainly in the last years, coinciding with the completion of most of PVV interventions, especially in slums S3 and S4. Distinctly, S2 presented a reduction of mortality rates in the year when intervention started but not as vigorous in the years after works were completed. In S5, where the intervention started later (only in 2011) no decreasing rate was observed during the studied period. S3 and S5 did not follow the same pattern before interventions started. However, a deep graphical drop can be observed in S3 after starting the interventions, but not in S5 (see Figure 3).

For each intervened slum, when considered separately, the relative risk analysis showed a significant protective effect on the homicide rates considering either time of observation or cumulative time of exposure to the completed intervention, except for S5. There was a decrease in homicide rates of 6%, 10%, 8% and 11% per year of observation and of 19%, 27%, 34% and 68% per year of exposure to completed intervention for S1, S2, S3 and S4, respectively (Table 3).

Regarding the effect of intervention on homicide rates, when comparing each intervened slum to the group of slums without interventions, the overall risk ratio in S1 showed a significant protective effect (RR = 0.71; 95%CI: 0.68–0.82). Reversely, in S2, S3 e S4 we encountered a significant high-risk effect, while in S5 no effect was observed (Figure 4).

## 4. Discussion

This study analyzed a comprehensive and participatory urban renewal project in slums in the 6th largest city in Brazil and its effects on homicide rates related to the built environment interventions distributed over 11 years of observation.

It confirms, as shown by the high rates of homicides in slums, that violence tends to be more prevalent in urban areas with higher population density, markedly unplanned and socially and spatially excluded due to poor connectivity, topographic difficulties, poor access to social services and a myriad of other factors intrinsically linked. Further worsening the scenario, there is the consequent fraying of the social organization of these excluded territories, marked by the invasion of drug trade forces—their enemies and allies, what generally produce conflicts—the beliefs and values of consumption and power, the disrespect for diversity, besides the insufficiency of a citizen safety system, widely present nowadays. Also, the possession of firearms, still uncontrolled in Brazil, can influence and increase homicide rates [6,7,38,39,40,41,42,43,44].

Besides, the high homicide rates encountered herein are compatible with rates observed in other slums in Brazil and other Latin American countries, denouncing the inequity also present in Belo Horizonte [5,10,11,38,39,40,41,42].

Despite many theories relating socioeconomic conditions to violent mortality rates, the complex network of factors that contribute to the cause-effect chain between violence and social conditions remains unclear. Studies have suggested that most violent deaths are linked to inequalities in income distribution, social cohesion and social capital, individual and collective social trust, and levels of investments in education, health care, and housing [18,25,38,39,45,46,47].

Factors related to violence can be grouped into four categories: (1) the structure of the built environment in cities, (2) the culture of masculinity, (3) drug trafficking and, (4) inefficient judicial systems that foster impunity [48].

Class, color, gender and age intolerance, social inequalities, lack of employment opportunities and urban segregation, paired with the culture of force and power, the local drug trade and the availability of firearms and widespread use of alcohol are also associated with violence in urban contexts [9,20,38,39,40,42,43]. Moreover, in areas in the city where wealth and extreme poverty cohabit, violence tends to occur more frequently, as is the case of most state capitals in Brazil, such as Belo Horizonte [11,20,39,40,41].

In this study, despite a downward trend observed in some slums even before interventions had started, a significant decrease of risk rate was observed in four out of five intervened slums according to the calendar year and mainly as the time of exposure to the completed interventions increased.

This evidence of a decrease in homicide rates in the calendar year preceding the interventions but also in the following years, further decreasing once the interventions were completed (Figure 3, Table 3), is not a finding which is repeated in the rest of Brazil nor in Minas Gerais (MG), state of which Belo Horizonte is the capital.

A study carried out in Brazil shows that from 2002 to 2012, Minas Gerais presented a 40% increase in homicide rates, despite some stability in some years of this period. Brazil also showed an increase, however small, of 2.1% [11,43,44] in its rate. The dissonance between the rates are most likely explained because Brazilian rates were influenced by decreases in the rates of the most populous states such as Rio de Janeiro and São Paulo, following the disarmament policy implemented between 2003 and 2007 [11,43].

In spite of the timid campaign for the disarmament in Belo Horizonte, the study shows a reduction in firearm mortality rates from 2004 to 2014 [44]. Despite these data, homicide rates continued to grow by about 4.0% a year, and showed an increase of about 16.7% in Brazil from 2010 to 2015 [11].

Findings from our study, used for comparisons, show that the homicide rates of the formal city, excluding slums’ data, are stable over the years, presenting an average rate of around 25.9/100,000 in the studied period. However, the rates of the city including the slums’ data dropped during the same period and the average rate was 39.5/100.00, most probably related to the decrease in slum rates.

In this period, Belo Horizonte was strongly influenced by a democratic and popular administration with policies of inclusion and social protection such as Vila Viva Project, Participatory Budgeting, Federal Government Income Transfer Program (Bolsa Família Program) and many others, which have increased qualified access to health, education, social assistance, culture and leisure policies [13,15].

Besides, there was no direct stimulus to policing policies in the slums such as the pacifying police units in Rio de Janeiro. In Belo Horizonte, priority was given to urbanizing slums and providing them with urban and social services. This investment in social and urban policies, such as Vila Viva Project, may have made it possible to indirectly increase security in the slums. Increased accessibility, as an example, may have allowed better mobility for policing in the slums, which may have favored the rate drop. Although unassertive, the disarmament statute might have influenced this drop as well.

Thus, homicide rates in Belo Horizonte and especially in the slums with Vila Viva interventions declined, differently from what happened either in Brazil or Minas Gerais state between 2002 and 2012 [11]. Furthermore, the sharpest decline in homicides in four of the five studied slums, especially towards the end of the completion of the intervention, may be revealing that the interventions might be influencing these declines, coupled with other policies and interventions that need to be better clarified in new research steps.

Comparing intervened slums to the comparison group, only one slum (S1), where interventions started earlier (2005) accumulating on average 3.7 years of exposure in 2012, presented a significant protective effect of the intervention (Figure 4).

This finding may corroborate to the hypothesis that impact on the homicide rates depends on time of exposure to the conclusion of the intervention. The other slums did not present the same results, perhaps because there was not enough exposure time to the concluded interventions to differentiate the intervened slums from the slums without intervention. Therefore, it may be plausible that it will be necessary to allow more time of exposure to the concluded interventions in order to reach significant reduction in homicide rates. 

Hence, this study represents the first investigation step in an attempt to link the hypotheses that the urban renewal of areas with built investments can promote direct or indirect impact on health, especially on drastic events such homicides typically clustered in vulnerable and areas.

Studies on impacts of social determinants of health are still insufficient in slums. There are also difficulties in analyzing how revitalization projects in vulnerable territories modify their physical and social tissue and health indicators. However, projects and evaluations of policies in this direction already indicate that physical and social environment modifications in slums influence the urban determinants, morbidity, inequality and social problems, such as homicides [16,28,30,31,33,34,41,43,44,45].

In this study it was not possible to discard a downward trend in the homicide mortality rates that seems to be related to the Vila Viva Project, especially in the S1, S3 and S4 slums, and which can be related to indirect and direct effects of the interventions.

As a sensitive indicator of the “urban tragedy” and abandonment by the State, as we know, for precarious conditions of habitability and lack of access to public policies of social protection and guarantee of civil rights [49], the risk of dying by homicide may be directly or indirectly influenced by urban renewal projects.This premise is supported both in the logic of Defensive Space and the Broken Windows theory and, fundamentally, in the logic of greater accessibility to goods and services and the inclusion of the slums in the formal city, opening new possibilities for slum dwellers, which may influence the fall in homicides in slums.

As previously said, many possible factors of slum re-urbanization projects may indirectly affect homicide rates [14,28,30,31,33,34,38,41,42,43]. Some of them are hereby listed: the transference of the residence’s secure ownership, modifications in the physical and social environment as an increase in leisure and coexistence spaces, a decrease in stress and other mental health indicators, an increase in feelings of belonging to a territory, and pride of being part of the transformed environment of enhanced social cohesion and collective efficacy.

This study has some limitations, which could be explained by the existing differences among the slums in the study of Vila Viva Project. More precisely are the characteristics of the social organization of these territories, even before the interventions. This topic may be further clarified in the next phases of this study, including a household survey.

One limitation is related to the nature of the secondary data and the difficulties to evaluate such complex health event and interventions. This is due to the scarcity of explanatory variables and absence of residual or subjacent variables that could be associated with the slums’ internal factors or even socio-organizational actions of the Vila Viva Project that could have produced differential estimates.

Secondly, this study covered the analyses of the urban-built interventions related to the construction of housing units and urban renewal composed by public facilities. Therefore, data restrictions imposes further analysis of the socio-organizational and land ownership regularization aspects as part of the PVV project.

The third limitation is the necessary caution when considering a health event such as the homicide. Its multiple and complex determinations intertwined in networks and dynamic processes among individuals and environments throughout time may result in a non-linear relation. Therefore, many other factors not measured here could have led to creating confusion, resulting in reduced or increased rates in the observed effect.

Last, due to the nature of the available data, it was impossible in this analysis to evaluate the independent effect of the multiple variables because of the collinearity of the time variables with the intervention, as well as to proceed with other more in-depth and multi-level analyses. Similarly, we cannot exclude a descending trend in homicide rates, which is independent from interventions. Since we argue that homicide rates depend on time of exposure to the concluded works and slums with not enough exposure time to the concluded interventions did not show similar results, it is plausible to consider allowing more time of exposure to the concluded interventions in order to reach significant intervention related reduction in the homicide rates.

## 5. Conclusions

Despite the limitations described, this study found a decreasing trend in homicide mortality following Vila Viva’s interventions. We also observed that analysis of the impact of complex interventions on relevant public health events of complex causality, such as homicides in vulnerable urban areas, is still neglected [18,47]. Additionally, assessment experiences of urban renewal projects on reduction of homicides are very scarce in Brazil and, in that sense, we have expanded knowledge by contributing to understanding how a comprehensive urban intervention in slums affects homicides.

Our findings herein described may strengthen the necessity of new analytical approaches, as well as new mixed designs, in order to confirm the relationship between slum renovation projects and homicide rate reduction.

Moreover, to explore the relationship of these findings with variables related to job market, education, access to “Bolsa Familia” (conditional cash transfer program), intra-city indexes of Human Development, is considered important.

Indeed, this analysis will be expanded in the near future through the collection of secondary data on mortality due to homicides up to the year 2016–2017, in addition to a household survey and a qualitative study providing a baseline for a posterior longitudinal follow-up. This may produce a consistent analytical design using a quasi-experimental method comparing slums with and without interventions, where similar controls or controls adjusted by known variables could be used, thus respecting the premise of experimental studies [27,28,50,51,52].

The on-going evaluation is being carried out in an interactive and reciprocal process with the Belo Horizonte city stakeholders, in order to recommend, with a scientific basis, the most appropriate public policies targeting vulnerable dwellers in urban areas.

## Figures and Tables

**Figure 1 ijerph-16-00154-f001:**
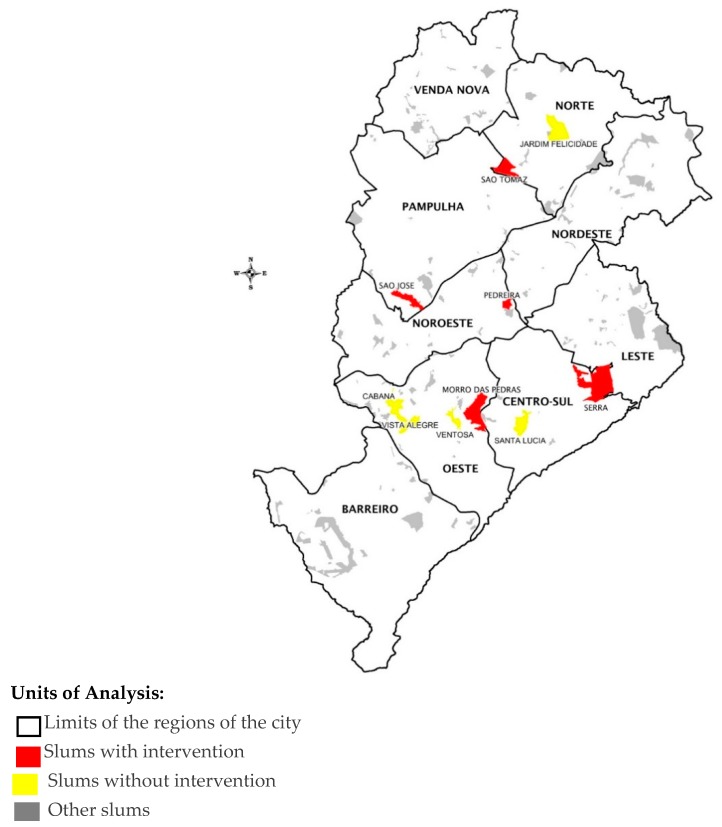
Belo Horizonte City—Limits of the regions, Slums with intervention, Slums without intervention and Other Slums.

**Figure 2 ijerph-16-00154-f002:**
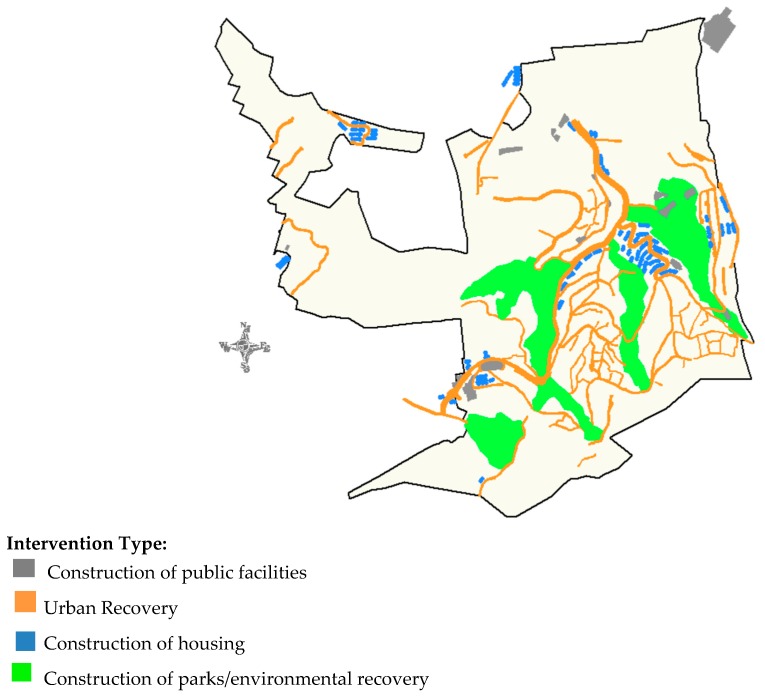
Vila Viva Project’s interventions by type of interventions—Vila S1 (Serra).

**Figure 3 ijerph-16-00154-f003:**
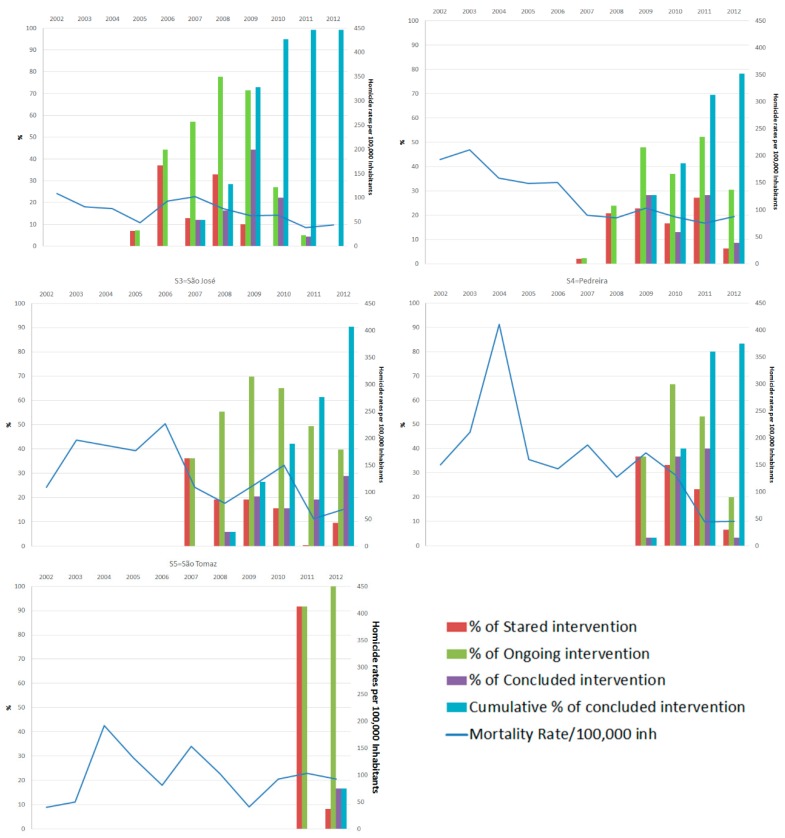
Timeline of Interventions and mortality rates of five slums in Belo Horizonte, 2002–12.

**Figure 4 ijerph-16-00154-f004:**
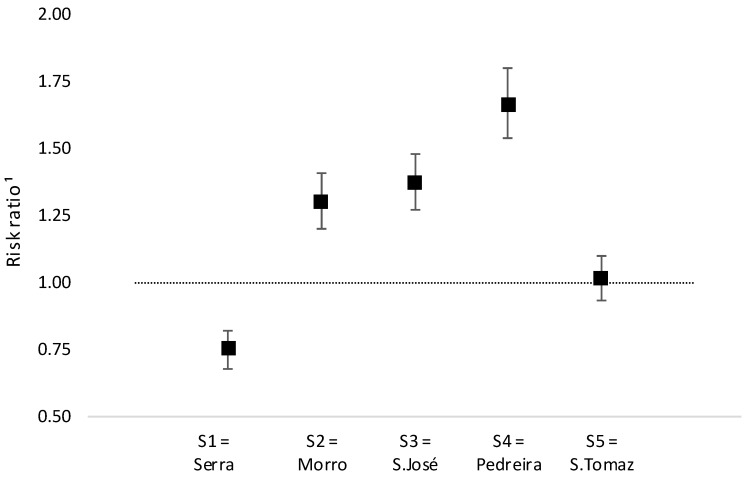
Risk ratio for homicides comparing intervened slums to grouped non-intervened slums. Legend: Poisson regression; reference: S0.

**Table 1 ijerph-16-00154-t001:** Characteristics of the studied slums according to demographic and social characteristics.

**Characteristics**	**Slums with Intervention—S1 to S5**					**Slums without Intervention—S0**				
	**S1 = Serra**	**S2 = Morro**	**S3 = S.José**	**S4 = Pedreira**	**S5 = S.Tomaz**	**Santa Lucia**	**Ventosa**	**Cabana**	**Vista Alegre**	**Felicidade**
**Total Population (*)**	43,299	19,070	7729	5570	10,147	14,522	8265	20,786	12,629	15,919
***% Population by sex (*)***										
Female	51.5	51.8	50.7	53	50.8	52	51	51.6	51.6	51.2
Male	48.5	48.2	49.2	47	49.1	47.9	48.9	48.4	48.3	48.8
***Population by age (*)***										
0 to 9 years old	22.7	22.2	25.0	21.1	21.0	24.2	21.9	20.7	18.5	19.3
10 to 19 years old	21.1	22.0	21.3	22.2	20.3	23.1	22.5	19.9	18.8	25.0
20 to 39 years old	34.4	34.1	34.7	33.2	36.1	34.0	34.4	35.7	36.2	33.0
40 to 59 years old	15.8	15.9	14.6	16.4	16.3	13.5	15.9	16.2	18.2	18.2
60 and above	5.9	5.8	4.4	7.0	6.4	5.1	5.4	7.4	8.3	4.5
***Population by race (**)***										
White	21.2	21.1	26.1	21.9	30.7	24.1	23.8	27.8	33.5	24.2
Black	77.1	78.0	72.2	77.8	67.7	74.5	74.8	70.3	65.6	73.3
Asian descent	1.4	0.8	1.6	0.3	1.5	1.3	1.3	1.6	0.7	2.3
Indigenous	0.3	0.1	0.1	0.0	0.1	0.1	0.1	0.3	0.2	0.2
***% of heads of household with income up to 2 minimum wages (*)***	89.7	100.0	94.1	99.8	76.7	96.3	98.7	97.4	68.1	68.5
***% residents by IVS (*)***										
Low	0.9	0.0	0.0	0.0	0.0	0.0	0.0	0.0	0.0	0.0
Medium	2.7	5.2	7.4	0.0	25.2	0.0	0.0	1.4	29.4	0.0
High and very high	96.8	94.9	92.8	100.0	75.1	100.0	100.0	98.9	70.7	100.0
**Characteristics**	**Slums with Intervention—S1 to S5**					**Slums without Intervention—S0**				
	**S1 = Serra**	**S2 = Morro**	**S3 = S.José**	**S4 = Pedreira**	**S5 = S.Tomaz**	**Santa Lucia**	**Ventosa**	**Cabana**	**Vista Alegre**	**Felicidade**
***Infrastructure and housing conditions***										
Inadequate sewage (***)	Yes	Yes	Yes	Yes	Yes	Yes	Yes	Yes	Yes	Yes
Inadequate water supply (***)	Yes	Yes	Yes	Yes	Yes	Yes	Yes	Yes	Yes	Yes
Inadequate garbage collection (***)	Yes	Yes	Yes	Yes	Yes	Yes	Yes	Yes	Yes	Yes
Urban drainage (****)	Yes	Yes	Yes	Yes	Yes	Yes	Yes	Yes	Yes	Yes
Geological risk of landslides/floods (***)	Yes	Yes	Yes	Yes	Yes	Yes	Yes	Yes	Yes	Yes
Insufficient road system and accessibility (***)	Yes	Yes	Yes	Yes	Yes	Yes	Yes	Yes	Yes	Yes
Poor housing conditions (***)	Yes	Yes	Yes	Yes	Yes	Yes	Yes	Yes	Yes	Yes
**Social issues—Violence (***)**	Yes	Yes	Yes	Yes	Yes	Yes	Yes	Yes	Yes	Yes
**Beginning of Occupation (***)**	1920–1940	1920–1940	1960–1970	1920–1940	1960–1970	1930–1950	1930–1950	1930–1950	1970	1970

Source: (*) Census 2000 [37]; (**) Census 2010 [37]; (***) URBEL-PBH [15,36] and BH-VIVA-OSUBH team.

**Table 2 ijerph-16-00154-t002:** Time of interventions and costs per works. Vila Viva Project, 2002 to 2012.

Timetable and Costs	Slums				
	S1 = Serra	S2 = Morro	S3 = S.José	S4 = Pedreira	S5 = S.Tomaz
Year of beginning of intervention	2005	2007	2007	2009	2011
Period of implementation of intervention (*)	2005 a 2011	2007 a ----	2007 a ----	2009 a ----	2011 a -----
Start Year of intervention completion	2007	2009	2008	2009	2012
Completion percentage in 2012	100	78.3	90.3	83.3	17
Total costs of intervention in dollars ($) (**)	$122,198,387.09	$49,782,258.06	$62,563,978.49	$22,910,752.68	$8,908,064.51
% of living expenses	70	45	85	55	45
% of expenditures with urban recovery works (***)	30	55	15	45	55

(*) the dashed line indicates that the works were still in execution in 2012. (**) does not include project costs, indemnities, consultancies, health facilities, education and social assistance. Conversion based on values of 2007 July. (***) including public facilities such as squares, parks and other sports and leisure areas Source: URBEL [38].

**Table 3 ijerph-16-00154-t003:** Homicide rates, risk ratios by calendar year and exposure time of intervention.

Variable	Intervened Slums				
	S1 = Serra	S2 = Morro	S3 = S.José	S4 = Pedreira	S5 = S.Tomaz
Homicide rates for the period (2002 to 2012; average per 100,000 inhabitants)	72.6	126.2	133.4	161.9	98.1
Cumulative time (years) of exposure to completed intervention in 2012 ^1^	3.7 years	2.2 years	1.9 years	1.6 years	0.5 year
	**Risk ratio (95% confidence interval) ^2^**				
Time of observation (calendar year: 2002 to 2012)	0.94 (0.91–0.96)	0.90 (0.88–0.91)	0.92 (0.91–0.94)	0.89 (0.88–0.90)	1.00 (0.98–1.02)
Cumulative time (years) of exposure to completed intervention	0.81 (0.76–0.87)	0.73 (0.67–0.79)	0.66 (0.61–0.72)	0.32 (0.26–0.38)	0.87 (0.54–1.38)

^1^ Computed from the first completed work; ^2^ Poisson regression.

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
