# Peer review of "Mortality from Homicides in Slums in the City of Belo Horizonte, Brazil: An Evaluation of the Impact of a Re-Urbanization Project"

_ijerph, 2019, doi:10.3390/ijerph16010154_

Round 1

Reviewer 1 Report

This research is very important and timely. As developing countries test different slum upgrading models most methods of evaluation still consider only economic impact in the model proposed by de Soto for titles. More attention is needed to the health and environmental impacts. T find the articles's discussion and conclusion well written and clear. Some further work is needed in the abstract, introduction, and methods for further detail. 

Abstract: the abstract must be more specific about the interventions.For instance, slum upgrading interventions, urbanization, land regularization, etc.. Authors could even cite the official name of the intervention. 

"Driven by a democratic and popular government and a process of participatory governance, in 2002 the Belo Horizonte City Hall (PBH, in Portuguese) started an extensive re-urbanization project of slums [12]." Is this participatory budgeting? 

For the Vila Viva project: where the vilas located in the same areas of the city? What is their geographical distribution? Also, for table 01, did the authors used field methods to collect data or is the data based on municipal evaluations available online for every item? I guess my questions is for urban drainage, for example, is the data available for each of the five vilas in the census? "Slum data collected by means of a PGE participation process were systemized based on

extensive documentary analysis by the BH-VIVA-OSUBH team" This needs further explanation.

Regarding the discussion: "This finding may corroborate to the hypothesis that impact of 37 the homicide rates is depending on time of exposure to the conclusion of the intervention." What about the impact of possible increases and decreases in policing nearby? Of course, police presence can impact homicide rates towards undesirable trends in slums. 

"Studies on impacts on social determinants of health are still insufficient in slums. There are also 47 difficulties in analyzing how revitalization projects in vulnerable territories modify their physical and 48 social tissue and health indicators.": Also lines 60-65 can be complemented with more research results on the impact of slum upgrading via participatory budgeting on health, especially declines in infant mortality:

Campbell, M., Escobar, O., Fenton, C., & Craig, P. (2018). The impact of participatory budgeting on health and wellbeing: a scoping review of evaluations. BMC public health, 18(1), 822.

Gonçalves, S. (2014). The effects of participatory budgeting on municipal expenditures and infant mortality in Brazil. World Development, 53, 94-110.

Touchton, M., & Wampler, B. (2014). Improving social well-being through new democratic institutions. Comparative Political Studies, 47(10), 1442-1469.

Walker, A. P. P. (2016). Self-help or public housing? Lessons from co-managed slum upgrading via participatory budget. Habitat International, 55, 58-66.

Author Response

Answers to Reviewer 1:

Firstly, I would like to sincerely thank you for the enormous contributions, especially for the articles provided on the evaluation of participatory budgeting and its benefits for community empowerment and improvement of health and wellness indicators. We read them and hope to incorporate participatory budgeting and its works into future analyses. Indeed, we incorporated them in the reference list of this paper.

 As to your very pertinent considerations, these are our comments, and many of them have been added to the text.

 The omission of information in the abstract is due to the standards of the journal in which it is under submission. But we hope to include your remarks by asking Ms. Cindy Cai.

1.     Abstract: the abstract must be more specific about the interventions. For instance, slum upgrading interventions, urbanization, land regularization, etc.. Authors could even cite the official name of the intervention. 

Thank you for your comment. The abstract-Background had changed to include the topics:

Background: Homicide rates in Brazil are one of the highest worldwide. Although not exclusive to large Brazilian cities, homicides find their most important determinants in cities’ slums. In the last decade, an urban renewal process has been initiated in the city of Belo Horizonte, in Brazil. Named  Vila Viva project, it includes structuring urban interventions such as urban renewal, social development actions and land regularization in the slums of the city. This study evaluates the project’s effect on homicide rates according to time and interventions.”

2.     "Driven by a democratic and popular government and a process of participatory governance, in 2002 the Belo Horizonte City Hall (PBH, in Portuguese) started an extensive re-urbanization project of slums [12]." Is this participatory budgeting? 

This is not directly the Participatory Budgeting, but a governance model that is built in a decentralized and participative way. The Vila Viva Project was developed under the management of the city of Belo Horizonte, based on technical analysis and from listening to the population in all participative forums, such as the old and currently extinct City Council, the Urban Policy City Council and also forums on Participatory Budgeting (PB). For the Vila Viva Project, in the existing forums, when the Global Specific Plan (PGE in Portuguese initials) was built.

And once again thank you, we will include the participative forums which the aforementioned articles refers to and other city forums, as well as the reference to Vila Viva and the PGE in this question which you pointed. Placed below:

This process includes participatory budget forums, sectoral and cross-sectoral  councils of urban and social policies, and participatory forums like those of Vila Viva Project, with similar models to what happens in other Brazilian cities[12-16]

Afonso, A.S .; Magalhães, M.C.F. Vila Viva Program: structural intervention in precarious settlements. Rev. Urban Habit. 2014, 1, 31-36, which seems quite adequate.

Azevedo,S.; Nabuco, AL. Democracia Participativa: a experiência de Belo Horizonte, 1ªed.; Editora Leitura: Belo Horizonte, Brasil, 2008; 292 p; ISBN 978-857358-844-6.

In addition, Vila Viva interventions are not chosen as a whole within the participatory budgeting. For being vast and structural, their costs (1, 2 Billion of Reais) for the current Vila Viva Projects (set in 12 slums) exceed the percentage available for them in the PB. But they often include interventions chosen by the PB.

3.     For the Vila Viva project: where the vilas located in the same areas of the city? What is their geographical distribution? Also, for table 01, did the authors used field methods to collect data or is the data based on municipal evaluations available online for every item? I guess my questions is for urban drainage, for example, is the data available for each of the five vilas in the census? "Slum data collected by means of a PGE participation process were systemized based on extensive documentary analysis by the BH-VIVA-OSUBH team" This needs further explanation.

Initially, I would like to say that the sources informed in table 01 were modified because they were mistaken. We inadvertently used the Mortality Information System as a source, which was also important for choosing the slums, and mainly for the comparisons, but it was not demonstrated in the table with the mortality rates. Thus, the correct sources are now described in table 01: (*) 2000 census; (**) 2010 census and (***) Urbel-PBH-OSUBH team.

As to the sources, the census shows inadequate supply of water and sewage and waste collection by households and their percentages, which was also useful to us.

However, our source was the analysis of documents, which are the PGEs of all slums, carried out by URBEL technicians in the urban and social field, with community participation.“The diagnosis is carried out by URBEL technicians in the urban and social field, with community participation. The purpose is to diagnose, in detail, the urban-environmental, socio-economic and legal scenario that addresses and allows the analysis and diagnosis of the following items: (1)degree of housing consolidation, based on the concentration of residents per household and the characteristics of the buildings; (2) degree of consolidation of the road system, identifying the physical characteristics and organization of local access; degree of water insalubrity, from the identification of critical drainage areas and the characteristics of local basic sanitation; (3) degree of geological-geotechnical consolidation, relating the types and levels of geotechnical hazards; (4)conditioning and restrictive characteristics of the occupation, through legal analysis and land condition analysis; and identification of the local social context, the issues experienced by the community and their access to several social policies (with their difficulties and virtues)[15, 36].”  
PGE data is not available online. It was delivered to us by URBEL. They were read and analyzed by the team of BH-VIVA-OSUBH. However, a thesis on the documentary analysis of 04 slums generated a publication which will be added to this article and is referenced here for your consultation: 
Silveira, DC, Carmo, R.F., Luz ZMP. O planejamento de quatro áreas do Programa Vila Viva na cidade de Belo Horizonte: uma análise documental. Cien Saude Colet  [internet periodical] (2017 / Jul). [Quoted on December 06, 2018]. Available at: http: //www.cienciaesaudecoletiva.com.br/artigos/o-planamento-de-quatro-areas-do-programa-vila-viva-n-de-belo-horizonte-an-analysis- documentary / 16299? id = 16299 
About the choice of slums: 
“Slum data collected through a participatory process carried out by the PGE and systematized by the BH-VIVA-OSUBH team’s extensive documentary reading and analysis of the slums' PGE in the regional and central archives of URBEL[36].Supplementary information of demographic indicators, as well as a compound index of social vulnerability were extracted, respectively, from the IBGE 2000 and 2010 censuses [37] and the City Health Department. Every information was georreferenced by census tract.In partnership with URBEL and with this information at hand, we tried to select slums with interventions scheduled to take place at different times, for current and future comparisons of the impact of interventions; slums with broader and more robust interventions; slums with and without interventions that besides being vast in their area and population size, were also situated near upscale neighborhoods; slums situated in less wealthy regions. Besides that, all slums selected were historical in the city. We also chose those with complex social and habitational issues, according to table 01, and  with similar characteristics.”Their location in the city is presented in the map that was added to the text, as figure 1 and also I have included in the text about the diagnosis and the step by step about the chosen slums.

4.     Regarding the discussion: "This finding may corroborate to the hypothesis that impact of 37 the homicide rates is depending on time of exposure to the conclusion of the intervention." What about the impact of possible increases and decreases in policing nearby? Of course, police presence can impact homicide rates towards undesirable trends in slums

Your question made me include a text in the discussion of the article, in the second paragraph, which refers to other determinants of death by homicide, such as the fraying of social organization aggravated by drug trade forces and the absence of a citizen safety system.Moreover, through comparisons with other studies carried out in Brazil, including possible improvements in the policing and the ones on disarmament policies, brought a new light to the discussion, so again I have included a new paragraph after the seventh paragraph of the discussion.               Unlike Rio de Janeiro, and despite the fact that violence is very present in the slums, the Vila Viva project in Belo Horizonte – taking into account the demands of the population – opted to offer services and policies and to stimulate social organization and cohesion instead of establishing of pacifying police units, which are known to have presented some distortions in Rio de Janeiro.It is certain that your opinion is absolutely right, because the very improvement of accessibility on public roads may have increased mobile policing. Perhaps even the new social organization may have provided negotiations for policing under the management of the state instead of the city. 
Below are the inclusions:In the second paragraph in the discussion:

“Furthermore, to worsen the scenario even more, there is the consequent fraying of the social organization of these excluded territories, marked by the invasion of forces of drug trade – its enemies and allies who generally produces  conflicts – the beliefs and values of consumption and power, the disrespect for diversity, besides the insufficiency of a citizen safety system very much present nowadays. Also, the possession of firearms, still uncontrolled in Brazil, can influence and increase homicide rates [6-7, 38-44].”

After the seventh paragraph in the discussion: 

“This evidence of a decrease in homicide rates in the calendar year preceding the interventions but also in the following years, further decreasing once the interventions were completed (figure 3, table 3), is not a finding which is repeated in the rest of Brazil nor in Minas Gerais (MG), state of which Belo Horizonte is the capital.

A study carried out in Brazil shows that from 2002 to 2012, Minas Gerais presented a 40% increase in homicide rates, despite some stability in some  years of this period. Brazil also showed an increase, however small, of 2.1% [11,43-44]. It grew less than the rate of MG most likely because its rates were influenced by decreases in the rates of most populous states such as Rio de Janeiro and São Paulo, following the disarmament policy between 2003 and 2007 [11, 43].

In spite of the timid campaign of disarmament in Belo Horizonte, the study shows a reduction in firearm mortality rates from 2004 to 2014[44].

Despite these data, homicide rates continued to grow by about 4.0% a year, and showed an increase of about 16.7% in Brazil from 2010 to 2015 [11].

Findings from our study, used for comparisons, show that the homicide rates of the formal city, without slums, are stable over the years, presenting an average rate of around 25.9/100,000 in the studied period. However, the rates of the city with the slums dropped during the same period and the average rate was 39.5/100.00, most probably related to the decrease in slum rates.

In this period, Belo Horizonte was strongly influenced by a democratic and popular administration with policies of inclusion and social protection such as Vila Viva, Participatory Budgeting, Federal Government Income Transfer Program(Bolsa Família Program) and many others, which have increased qualified access to health, education, social assistance, culture and leisure policies[13,15].

Besides, there was no direct stimulus to policing policies in the slums such as the pacifying police units in Rio de Janeiro. In Belo Horizonte, priority was given to urbanizing slums and offering them with urban and social services.

Notwithstanding, this investment in social and urban policies, such as Vila Viva, may have made it possible to indirectly increase security in the slums. Increased accessibility itself may have allowed an increased mobile policing in the slums, which may have favored the rate drop. Besides, even the disarmament statute might have influenced this drop as well. .

Thus, homicide rates in Belo Horizonte and especially in the slums with Vila Viva intervention declined, differently from what happened either in Brazil or Minas Gerais state between 2002 and 2012 [11]. Furthermore, the sharpest decline in homicides in four of the five favelas in the study, especially towards the end of the completion of the intervention, may be revealing that the interventions might be influencing these declines, coupled with other policies and interventions that need to be better clarified in new steps.”

However, this question is part of the limitation of the study at the present time, and which will require, in a new stage, crossings with new secondary data and analysis of the ongoing household survey.

5.     "Studies on impacts on social determinants of health are still insufficient in slums. There are also difficulties in analyzing how revitalization projects in vulnerable territories modify their physical and 48 social tissue and health indicators.": Also lines 60-65 can be complemented with more research results on the impact of slum upgrading via participatory budgeting on health, especially declines in infant mortality:

We thank you again and be certain that your comments have made us  reflect upon this and have changed and influenced the article in a positive way. I have included in the article the four references. 
References have been added and their numbers have changed. The revision of the English language was also done and I hope to be correct. 

Finally I would like to say that I added two small paragraphs in the introduction:

“For such urban disadvantage contexts and their avoidable and unjust consequences for the population, the urban and health conferences, studies, and revisions, government projects, international organizations and social movements recommend public urban and social policies, which are also intersectoral, structuring, and related to re-urbanization of townships and slums [20-24].”

“The international literature presents some conceptual models, different projects of urban interventions, as well as proposals for evaluations of urban revitalization policies that are associated with the reduction in criminality and other health indicadors[16,25-34]. Nevertheless, evaluation experience and even intervention projects are scarce in Brazil, where high homicide rates are ravaging, primarily in slums. “

I thank you once again.

Reviewer 2 Report

The strength of this work is bringing the potential devastating impact of serious inequities to light in a large Brazilian city. Unfortunately, even as the authors state, the comparisons are gross and the reduction of homicide rates is not necessarily related to the interventions. I find it difficult to think of even trying to make any comparisons here because of this issue.  There is no effort and again, as the authors state, indeed no way to find out what interventions may have contributed to the effects seen, even though it is likely they did so, at least in some way. 

Thus, I find it hard to support publishing this work in this format. The authors might consider rewriting the article to focus more as a description of the case, discussing possible relationships among potential variables to illuminate future research possibilities rather than relying on low power statistics to validate their claims. 

Author Response

Answer to the reviewer 2 :

First of all, we would like to thank you for reviewing this paper.

We agree regarding the remarks of the limits of the modeling adopted. However, as an extensive project this is the first part of a larger study named "BH-VIVA", with quantitative and qualitative methods, which is being developed in several phases, as mentioned in the article.

The new ways forward of this including ongoing analysis are: :

 - To explore the relationship  with variables related to the job market, education, access to “Bolsa Familia” (conditional cash transfer), intra-city indexes of Human Development;

- Add new  methods of statistical analysis such as time series analysis or spline method, already in progress;

- Extended the monitoring time for homicide rates and interventions until 2018 (in process);  

- A case study of homicides involving spatial analysis before and after the intervention, already in progress;

- Qualitative studies which go beyond the documentary analysis of all the slums with intervention, aiming to understand the implementation process and, in particular, the benefits that Vila Viva has brought to the residents, already in progress;

- A household survey and a systematic social observation study in a slum with intervention and a slum without intervention composed by self-perception and objective measurements, respectively,  of social and physical context, health and quality of life, comparing the slums before and after the intervention and object of future longitudinal studies in the slum without intervention, already in progress.

In view of the gravity of the homicides and of the lack of analyses of the impact of urbanization projects in Brazil with the goal of increasing opportunities for those living in vulnerable areas, reducing the disadvantages of this population and improving health indicators, we would once again ask the reviewer to consider:

- a methodological effort to choose the areas of study with regard to the documentary analyzes of each Global and Specific Plan of each slum and of census indicators for the selection of "intervened" and "non-intervened " slums with similar characteristics, important for the new quantitative and qualitative steps;

- the effort mainly to collect and systematize the calendar of interventions based on their scope and their different times in the slums that are the object of Vila Viva and its location in the space of the slums.

We would also ask you to consider as an initial step the analysis of the global plans and their compliance by the management and even without considering the possible confounding factors, to try and understand the effect of the interventions through the method used, even if it is less sophisticated and in the initial phase. However this early-stage method already indicates trends.

We aimed to seek ways and walked the step by step of a scientific research with the ultimate goal of producing information for the residents of the slums and city government agents.

And fundamentally, in a country with many early deaths and many potential years' worth of life lost to those excluded by an avoidable cause such as homicides, the search for evidence to support the implementation of policies that reduce inequalities and protect the residents of the slums.

Thank you for your attention and I hope to be graced with a positive evaluation.

Reviewer 3 Report

I liked the study, it made a compelling case and the findings are clearly important and should be published.  But there are questions.  I know from other work that Southern American homicides are proportionately more firearm related that in most other regions of the world, but there is no discussion of this, or the wider cultures of crime and violence, in the article.  This point relates to a further one, the urban development changes appear to be largely 'infrastructural re-urbanisation' whereas there is little comment on (I) what else is going on, (ii) policing or broader community safety policy changes, (iii) gun regulation.  You might say that this is not what the article was primarily investigating, and that is fair enough, but in the kind of criminological work with which I tend to be familiar, you would expect some commentary on this, in part to develop the explanation, in part to set the wider context (and it is in these areas that the qualitative work can help).  Even if nothing else significant was happening on these fronts, it would be useful to know that.  For instance, the reference in p. 2, para 3., to 'legalizing' the slums could be expanded a little.

Page 3: para 2 these is discussion of 'diagnosing the problem' - it would be helpful to know more about these diagnoses.

page 3: in the numbered points (point 3) there are some strange incomplete sentences

Relating to earlier point - p. 9, RESULTS: there is evidence of homicide trends falling before the urban interventions - this I not an unfamiliar phenomenon - but it would merit some discussion of what else was happening to influence these effects.

In some respects, I feel I'm critiquing the article for not being a different kind of study - which is unfair (it is what it is), and I do think it has merit and should be published but it does need a little tidying up and, in a few places, a little work on the English language.  For example....

p.2 para 3  're-urbanization models

para. 2.1.2  Vila Viva Project,   no 's

p. 6  para 2.1.4  every piece of information

section 4  Discussion

line 13  pungent,   prevalent

line 16  denouncing is the wrong word, I'm not quite sure what point is being made

line 35  non-intervened slums

38        is dependent

55    the whole reference to 'abandonment by the state' seems to point to rather more than just urban redevelopment, and wider context, about which we're told relatively little

73   this study only covered

100  Indeed sooner this ... will be   Indeed the sooner this analysis is

Author Response

First of all, I would like to thank you for the contributions and I believe that your remarks and suggestions are very thoughtful.  

1.      I liked the study, it made a compelling case and the findings are clearly important and should be published.  But there are questions.  I know from other work that Southern American homicides are proportionately more firearm related that in most other regions of the world, but there is no discussion of this, or the wider cultures of crime and violence, in the article.  This point relates to a further one, the urban development changes appear to be largely ‘infrastructural re-Urbanisation’; whereas there is little comment on (I) what else is going on, (ii) policing or broader community safety policy changes, (iii) gun regulation.  You might say that this is not what the article was primarily investigating, and that is fair enough, but in the kind of criminological work with which I tend to be familiar, you would expect some commentary on this, in part to develop the explanation, in part to set the wider context (and it is in these areas that the qualitative work can help).  Even if nothing else significant was happening on these fronts, it would be useful to know that. 

It is a very good point and we agree with the reviewer. Unfortunately, due to the length of the paper, we were not able to discuss in depth all the determinants associated to the high homicide rates in the city and its slums. But we discussed about some of them in this article.  

Although these determinants are scattered in the text (for example in the fifth paragraph), we added in the second paragraph of the discussion, after your recommendation, about  the drug trafficking and the deaths it produces. Also, we included  the issue of policing, which also produces deaths in its "war" against trafficking and militias, and is still considered very arbitrary in Brazil.   We are mentioning too ,  the possession of firearms and the lack of efficient policies for its control in Brazil, as described in the text below:

“Furthermore, to worsen the scenario even more, there is the consequent fraying of the social organization of these excluded territories, marked by the invasion of forces of drug trade – its enemies and allies who generally produces  conflicts – the beliefs and values of consumption and power, the disrespect for diversity, besides the insufficiency of a citizen safety system very much present nowadays. Also, the possession of firearms, still uncontrolled in Brazil, can influence and increase homicide rates [6-7, 38-44].”

Moreover, in accordance to the reviewer´s observations below on  the results (point 5), we explain better in the article how the rates dropped differently from the rest of Brazil.

Additionally, in another article submitted to  a Brazilian journal, we could describe the murders in Belo Horizonte City in the light of a comprehensive conceptual model that exposes the complex network of determinants in the city where this event is a synthesis of the existing inequality. Besides that, we contextualized a discussion  on the  growth of homicides in Brazil since the 1960s and especially in the 1980s to the  actual years of the study. Please, see below (point 5).

2.      For instance, the reference in p. 2, para 3., to ‘legalizing’ the slums could be expanded a little:

The process of land regularization is a consequence of the interventions proposed by the Vila Viva Program, it is planned to promote an integrated action that contemplates the legal, urban and social promotion aspects, with the participation of the community, aiming at the access of the local population property and decent housing, in order to ensure that public and private lands fulfill their social function. This component respects Municipal legislation, protocols of intentions between Federal , State and Municipal governments made possible by agreements to make municipal intervention feasible in these lands. Regarding private properties, it respects the Brazilian Civil Code.

The process is long and complex and finds difficulties for its complete implementation . I would like to clarify more but it is very extensive.

They follow the references but they are in Portuguese:

Afonso, A.S.; Magalhães, M.C.F. Programa Vila Viva: intervenção estrutural em assentamentos precários. Rev. Urban Habit. 2014, 1, 31-36.

Fernandes, E.; Pereira, H. D. Legalização das favelas: qual é o problema de Belo Horizonte? Planejamento e Políticas Públicas, n. 34, p. 173-99, 2010.

We have included a summary in the body of the article. Legalization of slums refers to interventions that promote the regularization of areas, through the transfer of ownership of land and property to the families that occupy them.

It was created in order to include slums in the formal city, legalize them (in the sense of legalizing the land and transferring property ownership), and improve living conditions and their residents’ quality of life.”

Reference in the article:

Afonso, A.S.; Magalhães, M.C.F. Programa Vila Viva: intervenção estrutural em assentamentos precários. Rev. Urban Habit. 2014, 1, 31-36.

3.      Page 3: para 2 these is discussion of ‘diagnosing the problem; - it would be helpful to know more about these diagnoses.

Indeed, we added more information to the diagnoses part, named Global Specific Plan for each slums (PGE in Brazilian acronym):

“The diagnosis is carried out by URBEL technicians in the urban and social field, with community participation. The purpose is to diagnose, in detail, the urban-environmental, socio-economic and legal scenario that addresses and allows the analysis and diagnosis of the following items: (1) degree of housing consolidation, based on the concentration of residents per household and the characteristics of the buildings; (2)degree of consolidation of the road system, identifying the physical characteristics and organization of local access; degree of water insalubrity, from the identification of critical drainage areas and the characteristics of local basic sanitation; (3) degree of geological-geotechnical consolidation, relating the types and levels of geotechnical hazards; conditioning and restrictive characteristics of the occupation, through legal analysis and land condition analysis; (4) and identification of the local social context, the issues experienced by the community and their access to several social policies (with their difficulties and virtues)[15, 36].” 
PGE data is not available online. It was delivered to us by URBEL. They were read and analyzed by the team of BH-VIVA-OSUBH. However, a thesis on the documentary analysis of 04 slums generated a publication which will be added to the article and is referenced here for your consultation: 
Silveira, DC, Carmo, R.F., Luz ZMP. O planejamento de quatro áreas do Programa Vila Viva na cidade de Belo Horizonte: uma análise documental. Cien Saude Colet  [internet periodical] (2017 / Jul). [Quoted on December 06, 2018]. Available at: http: //www.cienciaesaudecoletiva.com.br/artigos/o-planamento-de-quatro-areas-do-programa-vila-viva-n-de-belo-horizonte-an-analysis- documentary / 16299? id = 16299

4.      page 3: in the numbered points (point 3) there are some strange incomplete sentences

Thank you. We now corrected it, as follow:

“Social development projects, which include: the construction of public facilities, in order to provide health, educational and social assistance; the construction of recreational and cultural areas, sports facilities, parks and green spaces, aiming to achieve environmental recovery and leisure; job and income generation programs as well as projects to foster participatory mechanisms in the community.”

5.      Relating to earlier point - p. 9, RESULTS: there is evidence of homicide trends falling before the urban interventions - this I not an unfamiliar phenomenon - but it would merit some discussion of what else was happening to influence these effects.

As mentioned before, we included one text in article aiming to respond the reviewer observation after the seventh paragraph of the discussion. We discussed about the  direct and  indirect consequence of re-urbanization  projects. 

            “This evidence of a decrease in homicide rates in the calendar year preceding the interventions but also in the following years, further decreasing once the interventions were completed (figure 2, table 3), is not a finding which is repeated in the rest of Brazil nor in Minas Gerais (MG), state of which Belo Horizonte is the capital.

A study carried out in Brazil shows that from 2002 to 2012, Minas Gerais presented a 40% increase in homicide rates, despite some stability in some  years of this period. Brazil also showed an increase, however small, of 2.1% [11,43-44]. It grew less than the rate of MG most likely because its rates were influenced by decreases in the rates of most populous states such as Rio de Janeiro and São Paulo, following the disarmament policy between 2003 and 2007 [11, 43].

In spite of the timid campaign of disarmament in Belo Horizonte, the study shows a reduction in firearm mortality rates from 2004 to 2014[44].

Despite these data, homicide rates continued to grow by about 4.0% a year, and showed an increase of about 16.7% in Brazil from 2010 to 2015 [11].

Findings from our study, used for comparisons, show that the homicide rates of the formal city, without slums, are stable over the years, presenting an average rate of around 25.9/100,000 in the studied period. However, the rates of the city with the slums dropped during the same period and the average rate was 39.5/100.00, most probably related to the decrease in slum rates.

In this period, Belo Horizonte was strongly influenced by a democratic and popular administration with policies of inclusion and social protection such as Vila Viva, Participatory Budgeting, Federal Government Income Transfer Program(Bolsa Família Program) and many others, which have increased qualified access to health, education, social assistance, culture and leisure policies[13,15].

Besides, there was no direct stimulus to policing policies in the slums such as the pacifying police units in Rio de Janeiro. In Belo Horizonte, priority was given to urbanizing slums and offering them with urban and social services.

Notwithstanding, this investment in social and urban policies, such as Vila Viva, may have made it possible to indirectly increase security in the slums. Increased accessibility itself may have allowed an increased mobile policing in the slums, which may have favored the rate drop. Besides, even the disarmament statute might have influenced this drop as well. .

Thus, homicide rates in Belo Horizonte and especially in the slums with Vila Viva intervention declined, differently from what happened either in Brazil or Minas Gerais state between 2002 and 2012 [11]. Furthermore, the sharpest decline in homicides in four of the five favelas in the study, especially towards the end of the completion of the intervention, may be revealing that the interventions might be influencing these declines, coupled with other policies and interventions that need to be better clarified in new steps.”

1.      In some respects, I feel I’m critiquing the article for not being a different kind of study - which is unfair (it is what it is), and I do think it has merit and should be published but it does need a little tidying up and, in a few places, a little work on the English language.  For example....

As mentioned above, your notes were very fair and reflecting on all of them has been very valuable to us and to the article, which became much richer. We incorporated the English corrections and did the review.

About the question below:

the whole reference to ‘abandonment by the state’ seems to point to rather more than just urban redevelopment, and wider context, about which we’re told relatively little

Here, as mentioned before we could not extend much about the determinants but, as we know, these redevelopment projects can influence health indicators, including homicides, not only by access to better habitability conditions. They are very important, but more important is what can come as a direct or indirect consequence of these projects, as described in the paragraphs. We completed these two paragraphs with a little of what is the abandonment of the state as requested with the phrase:

As a sensitive indicator of the “urban tragedy” and abandonment by the State,

“as we know, for precarious conditions of habitability and lack of access to public policies of social protection and guarantee of civil rights ….”

This abandonment is partially linked to a paragraph which we added, as requested, information on the diagnosis of the PGE that deals with the urban, economic, social and legal context. In the study, as we said, it was not possible to deal with all contexts at this stage.

Finally, references have been added and their numbers have changed. The revision of the English language was also done and I hope to be correct. 

I would like to say that I added two small paragraphs in the introduction:

“For such urban disadvantage contexts and their avoidable and unjust consequences for the population, the urban and health conferences, studies, and revisions, government projects, international organizations and social movements recommend public urban and social policies, which are also intersectoral, structuring, and related to re-urbanization of townships and slums [20-24].”

“The international literature presents some conceptual models, different projects of urban interventions, as well as proposals for evaluations of urban revitalization policies that are associated with the reduction in criminality and other health indicadors[16,25-34]. Nevertheless, evaluation experience and even intervention projects are scarce in Brazil, where high homicide rates are ravaging, primarily in slums. “

I thank you once again.

Round 2

Reviewer 2 Report

This manuscript is improved in English language in key areas, including discussion of the findings. Some small English language edits are still needed.

Further, the additional discussion significantly improves this paper-thank you! I think the real contribution of this paper is as you say, as a first step in discussion of a complex intervention on health needs of people living in favelas.

Re: Table 1. I am concerned about the use of "race" and white, black, etc., and especially "yellow". Race is a construct and thus "ethnicity" is now the usual term used. Are there other, more precise  terms for "white" and "black" that you can use? I find the term "yellow" particularly stigmatizing. I think the title is better left as "asian descent".

Author Response

Answers:

First of all I would like to thank you for your care and to say that I was very pleased with this new perspective.

We did the English revision again and I hope the new version is better.

Regarding the issue raised about the race, which I agree, follows the response and the change in the table in relation to the Asian descent.

The concept of race stands as a historical construction that varies between countries.

In Brazil, this construction was marked by racial prejudice and the ideal of "Whitening" of the population, subjugating the blacks, as a portrait of the slave society.

It was also marked by the strong miscegenation of the Brazilian society where in research conducted in Brazil when it is asked about the origin the population places itself as of Brazilian origin. And after successive researches by the Brazilian Institute of Geography and Statistics (IBGE) the population recognizes itself  for its physical characteristics, also ignoring its origin. This is certainly marked by veiled prejudice is still present in Brazil, which makes it difficult for the population to recognize itself as Black and placing itself in the brown (or ‘pardo’, i.e., of mixed race/ color) category. Perhaps this question also poses itself to the population of Asian and Indigenous origin. But it is still little studied.

Studies advocate the need for blacks to maintain this dichotomy between browns and blacks since blacks are even more discriminated in the health system than browns. And you need to identify this physical appearance / origin bias.

Indigenous and Asian decent do not seem to be a question that generates controversy already putting these questions as being analyzed. However, the indigenous people are victims of extreme prejudice, as well as the minorities who immigrated to Brazil and who also made our history, like the Asians descents. This last one is still little studied and or visible, but exists.

The analyzes are already done as Indigenous and Asian descent , White, and Black (adding blacks and “browns” to approaches that do not require this dichotomy, as in the case of homicides where the Black population (browns and blacks has a risk of up to 3 times than whites.) In those as access to health care, as discussed above the dichotomy remains.

And so the controversy still sets in. Many doubts exist. Social movements as part of the Black Movement advocate for Black ethnicity.

Particularly I think that the IBGE (race / color) format does not contribute to the construction of identities or to the reduction of racial prejudices, even recognizing problems related to the measurement of ethnicity in multiethnic societies such as the Brazilian. I agree too much with ethnicity.

But by the hour, unfortunately, this is the census information in force in Brazil where the race is what is built and there are still no other names for whites and blacks as you pointed out.

Thank you very much again.

Best regards,

Maria Angélica de Salles Dias
